# Adversarial Examples Are Not Real Features

**Ang Li**[1*]   **Yifei Wang**[2*]   **Yiwen Guo**[3]   **Yisen Wang**[4,5†]

[1] School of Electronics Engineering and Computer Science, Peking University
[2] School of Mathematical Sciences, Peking University
[3] Independent Researcher
[4] National Key Lab of General Artificial Intelligence,
School of Intelligence Science and Technology, Peking University
[5] Institute for Artificial Intelligence, Peking University

## Abstract

The existence of adversarial examples has been a mystery for years and attracted much interest. A well-known theory by Ilyas et al. [19] explains adversarial vulnerability from a data perspective by showing that one can extract non-robust features from adversarial examples and these features alone are useful for classification. However, the explanation remains quite counter-intuitive since non-robust features are mostly noise features to humans. In this paper, we re-examine the theory from a larger context by incorporating multiple learning paradigms. Notably, we find that contrary to their good usefulness under supervised learning, non-robust features attain poor usefulness when transferred to other self-supervised learning paradigms, such as contrastive learning, masked image modeling, and diffusion models. It reveals that non-robust features are not really as useful as robust or natural features that enjoy good transferability between these paradigms. Meanwhile, for robustness, we also show that naturally trained encoders from robust features are largely non-robust under AutoAttack. Our cross-paradigm examination suggests that the non-robust features are not really useful but more like paradigm-wise shortcuts, and robust features alone might be insufficient to attain reliable model robustness. Code is available at https://github.com/PKU-ML/AdvNotRealFeatures.

## 1 Introduction

Alongside the human-level or even superior performance of Deep Neural Networks (DNNs) in various tasks [22, 14, 10], concerns on the existence of adversarial examples constantly rise, regarding them as main threats that fool DNN classifiers with invisible perturbations [34, 12, 24, 26]. Among many explanations of adversarial examples [11, 36, 31, 30, 44, 45, 48, 28, 2, 3], the robust and non-robust feature perspective developed by Ilyas et al. [19] has received wide attention. Compared to previous explanations that regard adversarial examples as "bugs" of neural networks (*i.e., the model*), they claim that adversarial examples stem from non-robust "features" in the inputs (*i.e., the data*). Specifically, they argue that each example is composed of human-perceptible *robust features* (invariant to attack) and human-imperceptible *non-robust features* (sensitive to attack) that are both *useful* for classification. Under adversarial attacks, non-robust features from other classes are injected into the adversarial examples and lead to misclassification. To verify this point, they show that datasets containing only non-robust features can attain good classification performance; and similarly, datasets containing robust features can attain good robustness even with natural training. Supported by these observations, the perspective has been widely accepted ever since and many subsequential works are built upon their framework [5, 20, 37].

---

*Equal Contribution.
†Corresponding Author: Yisen Wang (yisen.wang@pku.edu.cn).

37th Conference on Neural Information Processing Systems (NeurIPS 2023).

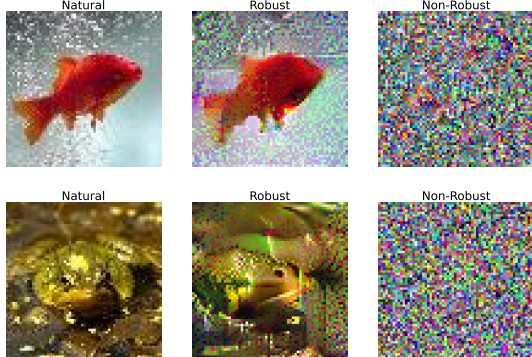

Figure 1: Tiny-ImageNet instances containing natural, robust, and non-robust features, respectively. The robust and non-robust instances are generated following the iterative optimization procedure in Ilyas et al. [19] from random noise. The robust features are semantically aligned to natural images, while the non-robust features are always noise-like.

Although both robust and non-robust features are shown to be useful for classification, the two still have large discrepancies, particularly in perceptibility. As shown by Ilyas et al. [19], robust features usually have rich semantic information such as distinguishable edges and meaningful combinations of colors; on the contrary, non-robust features are always noise-like and meaningless to humans, as shown in Figure 1. Therefore, it is natural for robust features to be useful, while the usefulness of noise-like non-robust features still seems suspicious and counter-intuitive. One may raise a natural question: *are non-robust features real (useful) features?*

In this paper, we aim to re-examine robust and non-robust features in a wider context in order to understand the true distinction between them. Particularly, we notice that a major limitation of Ilyas et al. [19] is the scoop of learning paradigms considered, as the usefulness of non-robust features is only evaluated on the supervised image classification. However, from the backdoor literature, we know that even a meaningless backdoor pattern can lead to the desired classification. A feature being useful for classification does not necessarily imply that it is truly useful. Therefore, it is reasonable to assume that truly useful features, *e.g.,* those utilized by humans, should work well for a wide range of learning paradigms instead of a single paradigm like the backdoor pattern.

Driven by the analysis, we take the first step by defining the *cross-paradigm* usefulness of robust and non-robust features, as a viable measure of true usefulness that generalizes previous definitions of Ilyas et al. [19]. A feature is defined to be truly useful only if it yields good representations across different learning paradigms. Aside from supervised learning, we further consider three modern self-supervised learning paradigms as the representatives: 1) contrastive learning that aligns augmented samples in the latent space, *e.g.,* SimCLR [27, 6, 15]; 2) masked image modeling that predicts the masked patches from the unmasked context, *e.g.,* MAE [9, 4, 16]; and 3) diffusion models that learn to restore images corrupted by Gaussian noise, *e.g.,* DDPM [17, 33]. Among the four paradigms, supervised learning and contrastive learning are discriminative tasks, while masked image modeling and diffusion models are generative tasks. Previous studies show that all these different paradigms yield representations with good downstream performance when pretraining on *natural images* [6, 16, 50]. Therefore, we would intuitively expect that if non-robust features are as useful as natural image features, we can also learn good representations from non-robust features alone. Similarly, we can define *cross-paradigm* robustness that evaluates whether a feature can yield robust representations (with robust prediction on downstream tasks) across different learning paradigms.

To validate their true usefulness, we follow the same procedure of Ilyas et al. [19] and construct robust (non-robust) datasets that only have robust (non-robust) features extracted from supervised models. To evaluate their cross-paradigm usefulness and robustness, we learn features with the four different learning paradigms on the constructed datasets and evaluate the learned features with a linear probing head for classification (a common evaluation protocol in self-supervised learning). Surprisingly, we find that there exist notable discrepancies in the cross-paradigm usefulness: on the three (transferred) self-supervised paradigms, robust features yield excellent performance that nearly matches natural features, while non-robust features yield much worse prediction that is hardly usable, in sharp contrast to its own good performance on supervised learning. It clearly conveys that **non-robust features**

are *not* as (cross-paradigm) useful as robust/natural features. Therefore, unlike robust or natural images, non-robust features do not contain much meaningful information that is truly useful across different paradigms. Intuitively speaking, non-robust features are more like a certain backdoor pattern of natural images subject to the chosen learning paradigm (*e.g.,* supervised classification). As paradigm-wise shortcuts, these features are essentially uninformative when examined with other learning paradigms. We further verify this point by showing that adversarial examples crafted under different paradigms can hardly transfer among each other. To this end, we conclude that robust features are truly useful features while non-robust features are not (in the cross-paradigm sense). Unlike the counter-intuitive explanation of Ilyas et al. [19], our result justifies the human intuition that the noise-like non-robust features do not really capture the essence of the images.

Furthermore, we also re-evaluate the robustness of robust and non-robust features in this way. Interestingly, contrary to the findings in Ilyas et al. [19] that natural training on the robust dataset produces a robust classifier, we find these so-called robust features hardly show robustness when learned with other learning paradigms. We also observe that the supervised learned classifier is non-robust under more reliable attacks like AutoAttack [7]. Thus, we arrive at the conclusion that **on real-world datasets, both robust and non-robust features extracted by Ilyas et al. [19] are actually non-robust, in both in-paradigm and cross-paradigm senses**. Although robust features are shown to exist in toy models when explicitly designed [39], there is by far no evidence that robust features exist in real-world datasets, at least not extractable.

To summarize, the main contributions of this work are three folds:

- In order to evaluate robust and non-robust features in a broader context, we generalize the notions on the usefulness and robustness of robust and non-robust features to a cross-paradigm sense. This perspective enables us to get rid of potential paradigm-wise shortcut effects and evaluate true feature informativeness.
- For usefulness, we find that robust features are both in-paradigm and cross-paradigm useful like natural features. Instead, non-robust features are only useful in-paradigm, and their usefulness dramatically degrades when transferred to other paradigms, suggesting that they are more like paradigm-wise shortcuts instead of real features. We further verify this point by showing that adversarial examples also have poor transferability across different paradigms.
- For robustness, we find that robustness obtained from the constructed robust dataset is actually a false sense of robustness when evaluated cross-paradigmly on more reliable attacks. The loss of this key evidence would raise the question of whether robust features really exist in real-world datasets.

Last but not least, although the main messages of this paper seem rather negative, this view also points out potential avenues to better adversarial robustness. In particular, the paradigm-wise behaviors of non-robust features suggest that it may be inadequate to perform adversarial training on a single learning paradigm, and a mixture of adversarial training on multiple paradigms may come to the aid. In the meantime, the non-robustness of robust dataset indicates a more comprehensive view of adversarial vulnerability: input data (or features) are *not* the only source of adversarial vulnerability, and only combating the vulnerabilities in both data and models can lead to true robustness.

## 2 A Cross-Paradigm View of Robust and Non-robust Features

### 2.1 Background

**Robust and Non-Robust Features.** As in Ilyas et al. [19], features are defined as a function mapping from input space to real numbers, namely $f : \mathcal{X} \to \mathcal{R}$. The robust and non-robust features are then described with the following definitions for a binary classification task:

- $\rho$-**useful features:** For a dataset $\mathcal{D}$, a feature $f$ is $\rho$-useful if the feature is correlated with the label:
$$\mathbb{E}_{(x,y)\sim\mathcal{D}}[y \cdot f(x)] \geq \rho. \tag{1}$$
- $\gamma$-**robustly useful features:** For a $\rho$-useful feature, it is regarded as $\gamma$-robustly useful if it remains useful under certain range of perturbation:
$$\mathbb{E}_{(x,y)\sim\mathcal{D}}\Big[ \inf_{\delta\in\Delta} y \cdot f(x + \delta)\Big] \geq \gamma. \tag{2}$$

- **Useful, non-robust features:** These feature are the ones that are $\rho$-useful features, but are not $\gamma$-robust features for any $\gamma \geq 0$.

**Construction of Robust and Non-Robust Datasets.** Given a classifier $C$ and a dataset $\mathcal{D}$, Ilyas et al. [19] propose to construct a dataset $\hat{\mathcal{D}}$ which satisfies:

$$\mathbb{E}_{(x,y)\sim\hat{\mathcal{D}}}[f(x) \cdot y] = \begin{cases} \mathbb{E}_{(x,y)\sim\mathcal{D}}[f(x) \cdot y] & \text{if } f \in F_C \\ 0 & \text{otherwise,} \end{cases} \tag{3}$$

where $F_C$ represents the set of features utilized by $C$. Denoting $g$ as the mapping from input $x$ to the representation layer in $C$, the new instance $x_r$ is obtained from $x$ through following optimization:

$$\min_{x_r} ||g(x) - g(x_r)||_2. \tag{4}$$

If the classifier $C$ (*e.g.,* ResNet-50) is adversarially trained, the constructed dataset $\hat{\mathcal{D}}$ is regarded as a *robust dataset*. As for a standardly trained classifier, the dataset $\hat{\mathcal{D}}$ is regarded as a *non-robust datatset*.

**Basic Conclusions of Ilyas et al. [19].** Utilizing the constructed datasets, Ilyas et al. [19] empirically show that models trained on the non-robust dataset are *useful*, *i.e.,* they can attain good classification performance on natural test data, comparable to models trained on the natural dataset. This supports their claim that non-robust features are sufficiently useful to obtain good generalization. This perspective well explains the *existence* of adversarial examples as well as their *transferability*, since adversarial examples contain non-robust features from the misclassifed classes that are useful for different models. Therefore, as they put it, adversarial examples are features, not bugs. Also, they also show that natural training on the robust dataset (containing robust features alone) can yield robust models. In this way, they view adversarial vulnerability purely from a data (or feature) perspective, regarding them as a result of different priors on extracted features.

**Self-Supervised Learning.** Aside from the supervised learning paradigm studied in Ilyas et al. [19], self-supervised learning (SSL) also received wide attention in recent years. Without manual labels, SSL methods utilize self-supervision to learn meaningful features from unlabeled data and have achieved impressive progress in recent years [6, 16, 17]. Generally speaking, an SSL algorithm pre-trains a feature encoder $f : \mathcal{X} \to \mathcal{Z}$, mapping from the input space to the latent space $\mathcal{Z}$. Afterward, the learned features are typically evaluated on the so-called linear probing task. Specifically, given a labeled dataset $(x, y) \sim \mathcal{D}$ (usually a subset of the pretraining data), we train a linear classifier $p : \mathcal{Z} \to \mathcal{Y}$ on top of learned features and use its classification accuracy (with the composed classifier $p \cdot f$) on the test data as a measure of the usefulness of learned features (representations).

## 2.2 Cross-Paradigm Notions of Robust and Non-Robust Features

In this part, we generalize the definitions of robust and non-robust features to a wider context beyond the supervised classification. For a rigorous discussion, we introduce some general definitions of the paradigm-wise feature usefulness and robustness and then define true usefulness and robustness in the cross-paradigm sense.

**Paradigm-Wise Definitions.** To facilitate features to generalize across different paradigms, we differ from Ilyas et al. [19] and adopt a more classic definition of features, filters in the input space, *i.e.,* $g : \mathcal{X} \to \mathcal{X}$. Note that common image features like textures, edges, and colors naturally fall into this category. We define a learning paradigm $T$ as a specific learning algorithm that learns a feature encoder $f$ with a loss function $L_T$ over a given dataset $\mathcal{D}$:

$$f_T = \arg\min_f \mathbb{E}_{x,y\in\mathcal{D}}\big[L_T(f(x), y)\big]. \tag{5}$$

Incorporating a feature $g$, the minimization problem can be further specified as

$$f_T^{(g)} = \arg\min_f \mathbb{E}_{x,y\in\mathcal{D}}\big[L_T(f(g(x)), y)\big]. \tag{6}$$

With each learning paradigm $T$, we can train an encoder $f_T^{(g)}$ for feature $g$. Then we evaluate the learned representations with a linear probing head $p$ on top of $f_T^{(g)}$ on $\mathcal{D}$. The usefulness of a feature $g$ relevant to the paradigm $T$ is defined as follows:

$$U(g, \mathcal{D}, T) = \max_{p\in\mathcal{P}} \mathbb{E}_{x,y\in\mathcal{D}}\mathbf{1}[p(f_T^{(g)}(x)) = y]. \tag{7}$$

That is, a feature $g$ is useful on paradigm $T$ if the linear probing head shows good classification performance with representations learned from the feature $g$ under the learning objective defined by the paradigm $T$.

Accordingly, the paradigm-wise feature robustness is defined as the remaining feature usefulness under local adversarial perturbations,

$$R(g, \mathcal{D}, T) = \max_{p \in \mathcal{P}} \mathbb{E}_{x,y \in \mathcal{D}} \max_{\|\delta\| \leq \varepsilon} \mathbf{1}[p(f_T^{(g)}(x + \delta_x)) = y]. \tag{8}$$

In other words, a feature is robust on paradigm $T$ if a standardly trained linear probing head shows good robustness with representations learned from the feature $g$ under $T$.

From this perspective, the conventional definitions of feature usefulness and robustness of Ilyas et al. [19] correspond to a special case of our definitions – when choosing the supervised learning task as the learning paradigm. In this way, the two processes, feature learning and feature evaluation, actually share the same learning objective, which may introduce spurious paradigm-wise shortcut effects. To get rid of this potential risk, we propose to re-define these concepts in a cross-paradigm sense.

**Cross-Paradigm Definitions.** Given a diverse set of learning paradigms $\mathcal{T} = \{T_1, T_2, \cdots, T_3\}$, we define a feature to be truly useful if it can generalize across multiple learning paradigms. Specifically, we define cross-paradigm usefulness as the worst paradigm-wise usefulness among these paradigms,

$$\text{(Cross-Paradigm Usefulness)} \ CU_{\mathcal{T}}(g, \mathcal{D}) = \min_{T \in \mathcal{T}} U(g, \mathcal{D}, T). \tag{9}$$

Similarly, we can define cross-paradigm robustness as the worst robustness

$$\text{(Cross-Paradigm Usefulness)} \ CR_{\mathcal{T}}(g, \mathcal{D}) = \min_{T \in \mathcal{T}} R(g, \mathcal{D}, T). \tag{10}$$

Nevertheless, since different paradigms usually attain performance on different levels, a direct comparison of their absolute performance may be unfair. In the worst case, if a certain paradigm has poor performance even when pretrained on the raw images, it would dominate other paradigms when computing cross-paradigm metrics, which, however, does not reflect the true feature usefulness. To mitigate this potential issue, we propose the paradigm-wise relative metrics as the ratio between the performance of the chosen feature and the performance of using all features (raw inputs),

$$\text{(Relative Usefulness)} \ RU(g, \mathcal{D}, T) = \frac{U(g, \mathcal{D}, T)}{U(\mathcal{D}, T)}, \tag{11}$$

$$\text{(Relative Robustness)} \ RR(g, \mathcal{D}, T) = \frac{R(g, \mathcal{D}, T)}{R(\mathcal{D}, T)}, \tag{12}$$

where given an encoder $f_T$ learned from the *raw images* under the paradigm $T$, we define

$$U(\mathcal{D}, T) := \max_{p \in \mathcal{P}} \mathbb{E}_{x,y \in \mathcal{D}} \mathbf{1}[p(f_T(x)) = y], R(\mathcal{D}, T) = \max_{p \in \mathcal{P}} \mathbb{E}_{x,y \in \mathcal{D}} \max_{\|\delta\| \leq \varepsilon} \mathbf{1}[p(f_T(x + \delta_x)) = y].$$

Accordingly, we can define the cross-paradigm relative usefulness and robustness as follows:

$$\text{(Cross-Paradigm Relative Usefulness)} \ CRU_{\mathcal{T}}(g, \mathcal{D}) = \min_{T \in \mathcal{T}} RU(g, \mathcal{D}, T), \tag{13}$$

$$\text{(Cross-Paradigm Relative Robustness)} \ CRR_{\mathcal{T}}(g, \mathcal{D}) = \min_{T \in \mathcal{T}} RR(g, \mathcal{D}, T). \tag{14}$$

## 3 Cross-Paradigm Usefulness of Robust and Non-robust Features

Built upon the evaluation framework established in Section 2.2, we first investigate the cross-paradigm usefulness of robust and non-robust features on real-world datasets in this section. With this generalized notion of feature usefulness, we are trying to answer the key question: *are robust and non-robust features truly useful?*

### 3.1 Setup

**Data Construction.** We include two commonly adopted datasets in our study, CIFAR10 [21] and Tiny-ImageNet-200 [49]. Aside from the raw images, following the same construction process in

Table 1: Evaluation of relative usefulness of robust features and non-robust features on four learning paradigms: MIM, CL, DM, and SL. The cross-paradigm relative usefulness is computed as the worst relative usefulness over the four paradigms. We also include the usefulness scores (evaluated only on the supervised task) reported in Ilyas et al. [19] for a comparison.

| Data | Feature | MIM | CL | DM | SL | Cross. Rel. Useful. | Ilyas et al. [19] |
|------|---------|-----|-----|-----|-----|---------------------|-------------------|
| CIFAR-10 | Robust | **0.896** | **0.791** | **0.831** | 0.880 | 0.791 | 0.854 |
| | Non-Robust | 0.307 | 0.433 | 0.512 | **0.914** | **0.307** | *0.823* |
| Tiny-ImageNet | Robust | 0.691 | 0.775 | 0.861 | 0.880 | 0.672 | 0.407 |
| | Non-Robust | 0.134 | 0.074 | 0.141 | 0.645 | **0.134** | *0.396* |

Ilyas et al. [19] (details in Appendix A), we further construct a robust version and a non-robust version for each dataset, which only contain robust and non-robust features, respectively.

**Learning Paradigms.** Besides supervised learning with the cross-entropy loss, we also include three self-supervised learning paradigms for a cross-paradigm evaluation: SimCLR [6] for Contrastive Learning (CL), MAE [16] for Masked Image Modeling (MIM), and DDPM [17] for Diffusion Models (DM). We then train linear probing heads for the encoders on the same dataset that it was trained on. The probing heads are directly applied to the output of the encoder for SimCLR and MAE models, while for DDPM model, since the U-Net encoder has complex high-dimensional hidden features, we use the global average pooled feature in the fourth upsampling layer in U-Net following Xiang et al. [50], which is shown to deliver excellent linear classification performance on par with other self-supervised learning paradigms. Also, we train a ResNet-50 on the datasets with Supervised Learning (SL) as our baseline. To summarize, the four learning paradigms considered in this work are $\mathcal{T} = \{\text{MIM}, \text{CL}, \text{DM}, \text{SL}\}$. After pretraining, we evaluate the classifiers' performance on the test sets of the two datasets. More details on training and evaluation are in Appendix A.

### 3.2 Non-robust Features are Not Cross-paradigmly Useful

We first examine the paradigm-wise usefulness of robust and non-robust features extracted from the supervised models. As shown in Table 1, models pretrained from robust datasets (containing only robust features) show comparable performance to those pretrained from the natural datasets (containing all features). In comparison, models pretrained from the non-robust datasets (containing only non-robust features) differ a lot across different paradigms: they work well for learning from the supervised task (ResNet-50), but much worse on all the other SSL paradigms (SimCLR, MAE, DDPM). It shows that in contrary to the perceptually aligned robust features that are useful for different paradigms, the noise-like non-robust features do not really contain meaningful information that is also useful for other SSL paradigms. Instead, these non-robust features extracted from supervised models only show usefulness on the supervised task. This sharp distinction suggests that non-robust features are more like paradigm-wise shortcut features rather than being truly useful.

For a comprehensive evaluation, we further compute the cross-paradigm *relative* usefulness following the definitions in Section 2.2. As shown in Table 1, the relative usefulness of non-robust features is significantly lower than that of robust features under various self-supervised learning paradigms. The overall cross-paradigm ends up being 0.307, which is much smaller than robust features (0.791). In comparison, these two kinds of features show similar performance (0.854 and 0.823) in Ilyas et al. [19] when only considering the supervised task. It shows that from a paradigm-wise view, the usefulness of features dramatically differs from the case where only supervised learning is considered. Again, it verifies that non-robust features are not as useful as natural and robust features when transferred to many other learning paradigms. For completeness, in Appendix C.1, we further examine the cross-paradigm transferability of the non-robust features extracted from the three SSL paradigms. Similarly, we find that these SSL non-robust features are non-transferable as well.

## 4 Cross-Paradigm Robustness of Robust Features

The results above challenge the claim from Ilyas et al. [19] that non-robust features are useful features by showing that non-robust features are not really useful in the cross-paradigm sense. Aside from the usefulness of non-robust features, another notable finding in Ilyas et al. [19] is that robust features

alone are enough for attaining model robustness. Specifically, they extract a robust version of a given dataset (*e.g.,* CIFAR-10) by distilling from a robust classifier such that the transformed samples only contain robust samples, and they show that a naturally trained classifier on the constructed "robust dataset" has good robustness. In this section, we further examine the robustness of the extracted robust features on different learning paradigms.

## 4.1  Setup

We construct the robust versions of CIFAR10 with a supervised $\ell_\infty$-robust ResNet-18 encoder [46] and a self-supervised $\ell_\infty$-robust ResNet-18 encoder [23], largely following the convention of Ilyas et al. [19]. We then train encoders on the robust datasets with the learning paradigms set $\mathcal{T} = \{\mathrm{MIM}, \mathrm{CL}, \mathrm{DM}, \mathrm{SL}\}$ and use the learned representations to naturally train a linear head. Finally, we evaluate the robustness of the composed classifiers (encoders + linear heads) with a modern reliable attack method, AutoAttack [7], under the perturbation budget of $4/255$, instead of the PGD attack [25] that may lead to over-estimated robustness [7]. More experimental details can be found in Appendix A.

## 4.2  Robust Features are Not Robust both In-Paradigm and Cross-Paradigm

We summarize the results in Table 2. Viewing from the in-paradigm results in Table 2a, we can observe that in supervised learning, PGD attacks can over-estimate the robustness of models trained from robust datasets, *e.g.,* $15.71$ (PGD) *v.s.* $3.65$ (AutoAttack). When evaluated under AutoAttack, robust datasets generated from either $\ell_2$-robust or $\ell_\infty$-robust models show poor robustness, being hard to match the robustness of adversarially trained source models. In other words, in practice, we believe it difficult to attain good robustness with only robust features (extracted following Ilyas et al. [19]). Their original conclusion on the good robustness of robust datasets could be due to the unreliability of PGD-based robust evaluation (also shown in the AutoAttack paper [7]). The severe over-estimation on robust datasets is potentially due to gradient obfuscation or masking [1] since these robust datasets are essentially generated with the PGD attack.

Extending the finding, we also evaluate the robustness of robust features on other self-supervised learning paradigms in Table 2b. Likewise, robust features also perform poorly and do not attain non-trivial robustness. Interestingly, when training on generative models like MIM and DM, robust features obtain slightly better robustness compared to the discriminative ones (CL and SL), even on natural data. Overall, the cross-paradigm robustness (defined in Section 2.2) is still bottlenecked by the SL itself. Based on these results, we conclude that robust features still cannot achieve model robustness alone (in either in-paradigm or cross-paradigm sense), without model-level interventions like adversarial training. It suggests that adversarial vulnerability may not come from the data alone, and a joint training strategy against both data-level and model-level vulnerabilities may be needed to attain real robustness. It worths noting that Tsilivis et al. [38] also found the non-robustness of the robust dataset of Ilyas et al. [19] in the supervised domain, while our results here give a full examination of robust features from both in-paradigm and cross-paradigm aspects.

## 5  Cross-Paradigm Transferability of Adversarial Attacks

The transferability across different models, even with different architectures [47, 40], is a fascinating fact of adversarial examples. The perspective of Ilyas et al. [19] provides a natural explanation for the phenomenon since non-robust features in adversarial examples are inherently useful such that they can affect the prediction of different models. Nevertheless, our studies in Section 3 reveal that non-robust features are only paradigm-wise but not cross-paradigm useful. It implies that adversarial transferability is also paradigm-wise. In other words, adversarial examples can hardly transfer between different paradigms. To verify the hypothesis above, we further investigate the cross-paradigm transferability of adversarial attacks in this section.

### 5.1  Cross-Paradigm Transferability of Attack Objectives

Since different paradigms mainly differ by their learning objectives, adversarial attacks *w.r.t.* different learning objectives (*e.g.,* SL, CL, MIM, DM) can generate adversarial examples of different paradigms. To study the transferability between different attack objectives, we consider two ResNet-18 backbone

networks (learned with SL and CL, respectively), attack them with an objective, and observe the change of both learning objectives.

**Setup.** We consider two objectives: InfoNCE loss [27] in CL and Cross Entropy (CE) loss in SL. Adversarial examples are generated with $l_{\inf}$ AutoAttack[7] bounded by $\varepsilon = 4/255$, using the default attack settings on CIFAR-10 (details in Appendix A). To be consistent, we compute the InfoNCE loss for a supervised encoder by appending a re-trained projection head on top of the fixed encoder. When computing the gradient during attacks, we also retain the standard data augmentations in SimCLR [6] and differentiate through these operators with the Kornia package [32]. For further ablation study on the influence of these components on transferability, see Appendix B.

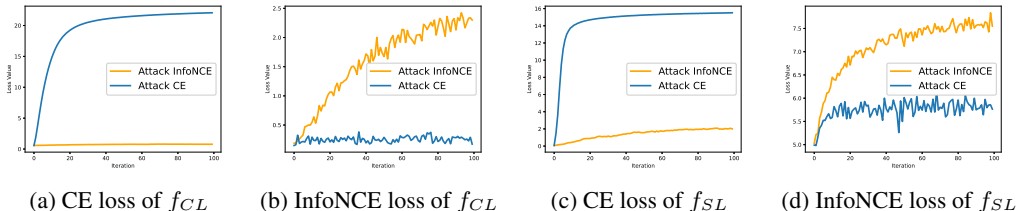

| (a) CE loss of $f_{CL}$ | (b) InfoNCE loss of $f_{CL}$ | (c) CE loss of $f_{SL}$ | (d) InfoNCE loss of $f_{SL}$ |

Figure 2: The change of loss value *v.s.* the attack iteration steps when using different attack objectives, CE loss (blue lines) or InfoNCE loss (orange lines), and backbones, trained by SL (Figure 2a & Figure 2b) or CL (Figure 2c & Figure 2d) from different paradigms.

**Results.** We plot the change of loss values during the attacking process in Figure 2, from which we can identify a general trend: there is a significant difference between the changes of different objectives, and maximizing one attack objective has a limited effect on the other objective. This distinction clearly implies that adversarial examples generated with attack objectives from different paradigms have poor transferability among each other.

## 5.2 Cross-Paradigm Transferability of Backbone Encoders

Aside from the attack objective discussed in Section 5.1, different paradigms also learn different paradigm-wise feature encoders that also have a large influence on the generation of adversarial examples. To further analyze the paradigm-wise influence of feature encoder on adversarial transferability, we generate adversarial examples with backbone encoders obtained by the four different paradigms, while keeping the attack objective to be the same CE loss (using the linear head learned for each paradigm). Besides the default ResNet-50(SL-RN) model, we further include three different

Table 2: Absolute robustness of robust features on four paradigms: MIM, CL, DM, and SL, on CIFAR-10. The robust features are extracted by pretrained models from two different source paradigms: supervised learning and contrastive learning. Different from Ilyas et al. [19] that use 1000-step PGD [25] for robust evaluation, we adopt a more reliable attack algorithm, AutoAttack [7].

(a) Robustness evaluation (with PGD and AutoAttack) for classifiers naturally trained with robust datasets (SL).

| Source Model | $\ell_2, \varepsilon = 0.5$ | | $\ell_\infty, \varepsilon = 4/255$ | |
| | PGD-1000 | AutoAttack | PGD-1000 | AutoAttack |
| --- | --- | --- | --- | --- |
| $\ell_2$-robust classifier | $15.71 \pm 0.21$ | $3.65 \pm 0.58$ | $6.36 \pm 1.66$ | $0.54 \pm 0.16$ |
| $\ell_\infty$-robust classifier | $13.96 \pm 0.12$ | $5.03 \pm 0.16$ | $17.63 \pm 0.16$ | $3.52 \pm 0.31$ |

(b) Cross-paradigm robustness (AutoAttack by default) of encoders naturally trained from natural or robust datasets. The robust datasets are generated from $\ell_\infty$-robust encoders trained with the SL or CL paradigm.

| Training Dataset | CL | MIM | DM | SL | Cross. Rob. |
| --- | --- | --- | --- | --- | --- |
| Natural | $0.22 \pm 0.03$ | $2.33 \pm 0.12$ | $3.41 \pm 0.32$ | $0.00 \pm 0.01$ | 0.00 |
| Robust w/ SL model | $0.23 \pm 0.01$ | $2.73 \pm 0.33$ | $2.54 \pm 0.20$ | $3.52 \pm 0.31$ | 0.23 |
| Robust w/ CL model | $1.28 \pm 0.01$ | $1.16 \pm 0.23$ | $1.19 \pm 0.14$ | $2.21 \pm 0.29$ | 1.16 |

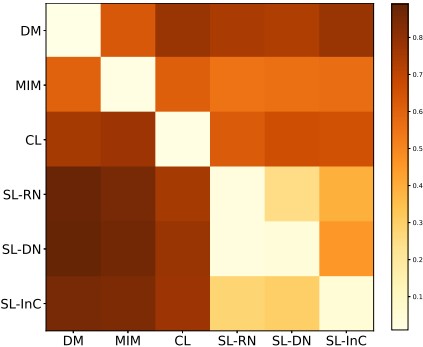

Figure 3: The cross-paradigm robustness of adversarial examples generated with encoders from different learning paradigms. The $(A, B)$-th cell represents the accuracy of adversarial examples generated with an $A$-paradigm model (encoder with linear head) when evaluated on a $B$-paradigm model (encoder with linear head). Darker colors (*i.e.,* higher accuracy) indicate worse transferability of adversarial examples.

supervised models, DenseNet-121 (SL-DN) [18], and Inception-V3 (SL-InC) [35] as baselines. We plot the transferred adversarial robustness in the confusion matrix in Figure 3.

The results in Figure 3 demonstrate two important messages: 1) adversarial examples transfer well across different architectures (SL-RN, SL-DN, SL-InC) under the same paradigm (SL), and 2) adversarial examples transfer poorly between different paradigms, even under the same architecture (for example, both CL and SL adopt ResNet-50 as backbones). It suggests that for adversarial transferability, paradigms have much more influence than model architectures. This phenomenon can be well explained by the paradigm-wise shortcut-like behaviors of non-robust features (Section 3), and it indicates that adversarial transferability is also largely a paradigm-wise phenomenon.

### 5.3  Relationship to Natural Transferability between Paradigms

In the discussions above, we demonstrate that adversarial examples do not have (good) transferability between different learning paradigms. We note that this discovery in *adversarial transferability* is not in contradiction to the good *natural transferability* between different paradigms. Indeed, many existing works [6, 15, 41, 16, 50, 52] show that representations learned from SSL have good downstream performance, particularly when evaluated with linear probing. Moreover, theoretical guarantees on the downstream classification performance have also been established for contrastive learning [13, 42, 43, 8], non-contrastive learning [53], and masked image modeling [51]. Since adversarial examples are essentially out-of-distribution examples (not drawn from the natural data distribution), the generalization guarantees on natural data do not apply. The fact that these paradigms have good natural transferability and poor adversarial transferability serves as another piece of evidence for our understanding that the misclassification of adversarial examples is caused by paradigm-wise shortcuts instead of real useful features.

## 6  Conclusion

In this paper, we have investigated the real usefulness and robustness of robust and non-robust from a wider context. By studying their behaviors across four different learning paradigms, we have found that robust features are as useful as natural features, while non-robust features generated by attacking supervised models become largely useless when transferred to other self-supervised learning paradigms, indicating that non-robust features are not real features but more like paradigm-wise shortcuts. Meanwhile, we have also shown that robust datasets containing only robust features are insufficient to attain model robustness under AutoAttack, indicating that feature non-robustness is not the only source of adversarial vulnerability. Posed as a challenge to common beliefs of robust and non-robust features, this work advocates the idea that their real usefulness and robustness should be examined under a cross-paradigm perspective as well as more reliable attacks.

## Acknowledgements

Yisen Wang was supported by National Key R&D Program of China (2022ZD0160304), National Natural Science Foundation of China (62006153, 62376010, 92370129), Open Research Projects of Zhejiang Lab (No. 2022RC0AB05), Beijing Nova Program (20230484344), and CCF-Baichuan-EB Fund.

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

# A  Experimental Details

Here we elaborate the experimental and technical details. First, we specify how the robust/non-robust datasets are constructed in our experiments. Then we detail the pre-training and linear probing hyper-parameters. Finally, we illustrate the settings of our robustness evaluation framework.

## A.1  Dataset Construction

We largely follow Ilyas et al. [19] in the process of dataset construction. Unless specified, we use the robust and non-robust datasets generated from images for experiments (Section 3, Section 5). In Section 4, we generate the robust CIFAR10 with both supervisedly trained and contrastively trained ResNet-18. All constructed datasets are obtained via $\ell_2$ PGD optimization. The detailed settings are listed below

Table 3: Experimental configurations in dataset construction. We generate robust and non-robust CIFAR10 starting from images (initialized with randomly drawn different images in the dataset) or from noise (random Gaussian noise).

| Dataset | Iteration | Step Size | $\varepsilon$-Cons. |
|---|---|---|---|
| Robust CIFAR10 (From Image) | 1000 | 1.0 | $\infty$ |
| Robust CIFAR10 (From Noise) | 10000 | 1.0 | $\infty$ |
| Non-Robust CIFAR10 (From Image) | 1000 | 0.1 | 0.5 |
| Non-Robust CIFAR10 (From Noise) | 10000 | 0.1 | 0.5 |
| Robust Tiny-ImageNet-200 (From Image) | 2000 | 1.0 | $\infty$ |
| Non-Robust Tiny-ImageNet-200 (From Image) | 2000 | 0.1 | 0.5 |

## A.2  Pre-Training

For evaluating the usefulness and robustness of features (Section 3, Section 4), we use ResNet-18 [14] for SL, MAE [16] backboned by ViT-t [10] for MIM, ResNet-18 for CL, and DDPM [17] backbone by UNet [29] for DM in all of the three versions of CIFAR10 [21] and Tiny-ImageNet-200 [49]. We additionally use supervisedly trained ResNet-50, DenseNet-50 [18], and InceptionV3 [35] for discussion in the paradigm-wise transferability of adversarial examples (Section 5). We list the hyper-parameter configurations below. We pick the default settings commonly adopted by the community when training the models and feel it unnecessary to intentionally optimize the hyper-parameters since our main research goal is focused on rethinking the essence of non-robust features.

Table 4: The hyper-parameter settings of pre-training on the three versions of CIFAR10.

| Paradigm | Model | Epoch | B-Size | LR | LR Sche. | W-Decay | Data Aug. |
|---|---|---|---|---|---|---|---|
| SL | ResNet-18 | 50 | 128 | 1e-1 | CosineAnnealingLR | 5e-4 | ✓ |
| | ResNet-50 | 50 | 128 | 1e-1 | CosineAnnealingLR | 5e-4 | ✓ |
| | DenseNet-50 | 50 | 128 | 1e-1 | CosineAnnealingLR | 5e-4 | ✓ |
| | InceptionV3 | 50 | 128 | 1e-1 | CosineAnnealingLR | 5e-4 | ✓ |
| CL | ResNet-18 | 2000 | 512 | 1e-4 | CosineAnnealingLR | 1e-5 | ✓ |
| MIM | MAE(ViT-t) | 2000 | 512 | 2e-4 | LambdaLR | 5e-2 | ✓ |
| DM | DDPM(UNet) | 1000 | 128 | 2e-4 | LambdaLR | 0 | ✓ |

## A.3  Linear Probing

We use linear probing to evaluate the representation learned by our encoders and we attach the trained linear heads to the encoders to transform them into classifiers. (Section 3, Section 4, Section 5).

Table 5: The hyper-parameter settings of pre-training on the three versions of Tiny-ImageNet-200.

| Paradigm | Model | Epoch | B-Size | LR | LR Sche. | W-Decay | Data Aug. |
|---|---|---|---|---|---|---|---|
| SL | ResNet-18 | 100 | 128 | 1e-2 | CosineAnnealingLR | 1e-5 | ✓ |
| CL | ResNet-18 | 1000 | 512 | 2e-3 | CosineAnnealingLR | 1e-5 | ✓ |
| MIM | MAE(ViT-b) | 1000 | 512 | 2e-4 | LambdaLR | 5e-2 | ✓ |
| DM | DDPM(UNet) | 1500 | 128 | 2e-5 | LambdaLR | 0 | ✓ |

Table 6: Hyper-parameter configuration of linear probing.

| Dataset | Epoch | B-Size | LR | LR Sche | W-Decay | Data Aug. |
|---|---|---|---|---|---|---|
| CIFAR10 | 30 | 256 | 0.1 | CosineAnnealingLR | 5e-4 | ✓ |
| Tiny-ImageNet-200 | 50 | 256 | 0.01 | CosineAnnealingLR | 1e-5 | ✓ |

## A.4 Robustness Evaluation

To replicate the results reported in Ilyas et al. [19], we use $\ell_\infty$ PGD-1000 [25] bounded by $\varepsilon = 4/255$ and $\ell_2$ PGD-1000 bounded by $\varepsilon = 0.5$ as baseline evaluation. Besides, we use $\ell_\infty$ AutoAttack bounded by $\varepsilon = 4/255$ for more reliable evaluation. Due to the randomness induced in the encoding process of MAE and DDPM, we use the randomized version of AutoAttack when evaluating their robustness, otherwise, we use the standard version of AutoAttack.

# B   Ablation Study on Cross-Paradigm Transferability

In Section 5.1, we have shown that adversarial examples generated by attacking contrastive learning (CL) have poor transferability on attacking supervised learning (SL). To further investigate how this happens, we conduct an experiment that gradually ablates each part of the InfoNCE loss.

Specifically, we notice three major differences when computing the CL and SL objectives: 1) CL usually adopts heavy data augmentations while SL does not; 2) CL adopts a projector head after the embedding while SL does not; and 3) CL and SL adopt different objectives: InfoNCE loss and CE loss. As shown in Table 7, ablating the augmentation and/or projector indeed brings a better attack success rate (lower robustness), since it bridges the differences between two paradigms. Following the same vein, we further consider ablating the InfoNCE loss with a simple alignment loss between the natural example $x$ and adversarial example $x'$, *i.e.,*

$$x_{\text{adv}} = \arg\max_{x'} \|f(x) - f(x')\|^2. \tag{15}$$

We can see from Table 7 that this alignment-only attacking objective obtains significantly lower robustness, though the difference to attacking CE is still large. This ablation experiment shows that the distinction between paradigms is not absolute, and we can improve their transferability by gradually eliminating their gap. We leave a more comprehensive study on this aspect to future work.

Table 7: Ablation study of model components when transferring adversarial examples generated *w.r.t.* the CL objective to the classification task on CIFAR-10.

| Paradigm | Attack Configuration | Robustness (%) |
|---|---|---|
| CL | No Attack | 78.51 |
| | InfoNCE loss + Projector + Augmentation (default) | 70.31 |
| | InfoNCE loss + Projector | 70.21 |
| | InfoNCE loss + Augmentation | 71.09 |
| | InfoNCE loss | 59.76 |
| | Alignment-only loss | 33.20 |
| SL | CE | 0.78 |

# C  Cross-Paradigm Transferability of Non-robust features from SSL Paradigms

In the main paper, we have mainly examined non-robust features extracted from an SL (supervised learning) model and show their non-transferability in a cross-paradigm sense. For completeness, we further examine non-robust features generated from the other three SSL paradigms considered in our work: CL, MIM, and DM. We follow the same hyperparameter setting as Ilyas et al. [19] for constructing non-robust dataset (Section 2.1) and for each paradigm, we generate adversarial examples by optimizing the perturbation to maximize the SSL pretraining objective *per se* (mode details below in Appendix C.1).

Table 8: Cross-paradigm transferability of non-robust features generated from SSL paradigms. Here the non-robust datasets are generated with noise initialization following the setting of Ilyas et al. [19].

| Accuracy | SL | CL | MIM | DM |
|---|---|---|---|---|
| CL-non-robust | 18.71 | 60.02 | 10.09 | 13.66 |
| MIM-non-robust | 12.02 | 21.65 | 22.19 | 15.47 |
| DM-non-robust | 14.07 | 14.11 | 18.07 | 31.08 |

Table 9: Here the non-robust datasets are generated with noise initialization following the setting of Ilyas et al. [19]. Here the non-robust datasets are generated with random image initialization following the setting of Ilyas et al. [19].

| Accuracy | SL | CL | MIM | DM |
|---|---|---|---|---|
| CL-non-robust | 22.18 | 41.80 | 14.33 | 11.91 |
| MIM-non-robust | 14.11 | 9.29 | 32.80 | 21.78 |
| DM-non-robust | 12.47 | 9.86 | 18.25 | 39.41 |

Results are shown in Table 8 (with random noise initialization) and Table 9 (with random image initialization). We still observe the same trend as observed in the SL-induced non-robust dataset (Section 3), where non-robust features show much better usefulness in-paradigm compared to that of other paradigms. Thus, we can conclude that the cross-paradigm non-transferability still holds for non-robust features pretrained from SSL paradigms.

## C.1  Implementation Details

**Constrastive Learning:**  We leverage the non-robust features hidden in CL models by aligning the adversarial examples with the clean images. Specifically, the adversarial examples are obtained via the following optimization

$$\text{CL: } \delta_i^* = \arg\min_{\delta_i} \log \frac{\exp(f(\delta_i)/\tau)\exp(f(x_i)/\tau)}{\sum_{j=1,j\neq i}^{n} \exp(f(\delta_j)/\tau)\exp f(x_j)/\tau} \tag{16}$$

where $\delta_{i=1}^{n}$ denote the $n$ adversarial examples, $x_{i=1}^{n}$ denote the $n$ clean image, i.e., the attack target within a batch and $f$ is the encoder being attacked. The non-robust features generated this way look like random noise but are semantically close to the clean images for the model, which qualifies them for the definition of adversarial examples.

**Masked Image Modeling & Diffusion Model:**  The non-robust features for MIM and DM are obtained by maximizing the possibility of recovering the target (the clean image) from the adversarial examples. Denoting the procedure of random masking in MIM and adding Gaussian noise in DM as $\Gamma$, we can express the adversarial optimization of the two paradigms as follows

$$\text{MIM: } \delta^* = \arg\min_{\delta} ||f(\Gamma(\delta)) - x||_2^2 \tag{17}$$

$$\text{DM: } \delta^* = \arg\min_{\delta} ||f(\Gamma(\delta)) - (x - \delta)||_2^2 \tag{18}$$

with $f$ denoting the model and $x$ denoting the clean image. Conceptually, we assume the non-robust features of MIM to be invisible perturbations that contain information for the model to reconstruct the clean image from it. With the diffusion models predicting noise added to the input, the non-robust features of DM are similarly defined as small perturbations fooling the model to predict the difference between the clean images and the adversarial examples. Note that we fix the timestep used in generating the adversarial examples to be the same as the one used in turning the DM into classifiers.

