# OpenReview forum: "Adversarial Examples Are Not Real Features"
_NeurIPS.cc/2023/Conference — NeurIPS 2023 poster_

### Official Review · Reviewer_NVbH · 2023-07-02

**Soundness:** 3 good
**Presentation:** 2 fair
**Contribution:** 3 good
**Rating:** 7
**Confidence:** 4

**Summary:**

This paper builds upon the work of [12] "Adversarial examples are not bugs, they are features" written by Ilyas et al. The authors of [12] introduced the concept of robust and non-robust features. According to [12], non-robust features alone can be useful for the classification task. The authors of the current paper show that such features however are not extendable to different tasks. They show this through linear probing and auto encoders. Furthermore, according to the authors of [12], robust features alone are able to create a robust dataset. However, the findings of this paper contradicts this notion.



**Strengths:**

1. This paper extends the definition of what means for a feature to be robust vs non-robust. Instead of relying on a single classification task, they extend the definition to a task wise perspective. This allows them to define "Absolute Useful" and "Relative Useful" features. Intuitively relative usefulness represents the importance of a feature more than absolute usefulness. Using these definitions, the authors then define $\rho$-useful and $\gamma$-useful features.
2. The authors then ask a series of compelling questions. They try to find the answer to these questions empirically. In the first experiment they find that using the task definition, non-robust features perform poorly (even though not non-negligible). This indicates that non-robust features are actually not useful. Next they find that the essence of non-robust features are task-specific and they should not be considered as real features. The next question is the most interesting one. The authors find that robust features alone are not enough to for robust training. This directly contradicts with the finding in [12]. Finally, they show that different tasks capture different non-robust features. These findings together shed more light on the nature of adversarial examples and the features related to them.

**Weaknesses:**

1. Even though the paper is pretty well written, there are some weakness in the presentation. These include several typos e.g., in (13) and (14) I believe the function $U(g, D, T)$ is mentioned incorrectly instead of $R(g, D, T)$. Similarly, in line 104 Figure 3 is mentioned, but I think that the authors meant Figure 2. Adding to these, I could not find a mention of Figure 1 in the text. This means that the evaluation framework is not properly explained. Some similar other details are also missing reducing the readability of the manuscript.

**Questions:**

1. In Figure 2, the clean dataset always performs better than the rest. Why is this the case always?
2. What are the authors intuition behind non-negligible features having non-negligible performance in Figure 2? Does this mean that some of this features can be transferred across some specific tasks?

**Limitations:**

1. As mentioned above the presentation of this paper can be improved. In my opinion doing so would elevate the paper further.
2. The authors ask four questions in the manuscript and answer them through experiments. It would be useful to mention these questions in the intro and comment on the usefulness of each of them.

---

> ### Author Rebuttal · Authors · 2023-08-09
>
> #### Title: Response to reviewer NVbH
>
> Dear Reviewer NVbH,
>
> Thanks for your careful reading and we are glad to hear that you appreciate the novelty and insights of this work. Below, we address your main concerns.
>
> ---
>
> **Q1.** Some suggestions regarding the presentation.
>
> **A1.** Thanks for your careful reading. We will fix these issues in the revision.
>
> ---
> **Q2.** I could not find a mention of Figure 1 in the text. This means that the evaluation framework is not properly explained.
>
> **A2.** Sorry for the confusion. We will add more descriptions of Figure 1 to explain the evaluation famework. Below is an outline:
> - First, we train a feature extractors $f_{T}$ with different kinds of tasks $T$ (SSL and SL ones) on a specific dataset;
> - Second, we perform linear probing by training a linear classification head on top of frozen features using labeled data;
> - Third, we evaluate the accuracy and robustness of the composed classifier (feature extractor + linear head) on test data.
>
> ---
> **Q3.** In Figure 2, the clean dataset always performs better than the rest. Why is this the case always?
>
> **A2.** We note that this phenomeon is similar in Ilyas et al's experiments (except that non-robust features perform much worse in ours). There could be two main reasons. **First**, the clean dataset contains both robust and non-robust features, which is supposed to provide more information for classification. **Second**, different from clean features using the original images, the (non-)robust dataset is constrcuted through iterative optimization from random inputs using an imperfect NN classifer. This process will inevitably lose information and introduce some noises to the constructed inputs, leading to performance degradation to some extent.
>
> ---
> **Q4.** What are the authors intuition behind non-negligible features having non-negligible performance in Figure 2? Does this mean that some of this features can be transferred across some specific tasks?
>
> **A3.** Indeed, it suggests that the human perceptible (non-negligible) features have good transferability across tasks, while human inperceptible features do not. Previously, Ilyas et al argue that despite the fact robust features are perceptible and non-robust features are not, non-robust features are still useful for classification. On the contrary, we show that their counter-intuitive understanding is questionable to some extent, as these non-robust features are largely useless on SSL tasks where natural and robust features work well. It indicates a fundamental difference between robust and non-robust features and only those human perceptible (robust) features are truly useful features, which justifies the rationality of human perception instead.
>
> ---
> Please let us know if you have further concerns/questions.

---

> > ### Comment · Reviewer_NVbH · 2023-08-17
> > **Response to author rebuttal**
> >
> > I thank the authors for taking the time to address the reviewers' concerns. In particular the addition of the TinyImageNet-200 results was very helpful. I would encourage the authors to make those results a part of the main manuscript. After reading the other reviews and the rebuttals I have decided to update my rating.

---

### Official Review · Reviewer_nwrB · 2023-07-05

**Soundness:** 2 fair
**Presentation:** 1 poor
**Contribution:** 2 fair
**Rating:** 5
**Confidence:** 4

**Summary:**

This work analyzes the robustness of deep networks under various tasks, analyzes how adversarial attacks between models trained on different tasks transfers between them, and does so through a newly proposed framework.

**Strengths:**

This paper delves into the robustness of models trained under different tasks (SSL training) and runs thorough analysis on CIFAR10 datasets to compare these models.

As SSL trained models continue to be used, this paper's research direction is interesting and useful.

The newly proposed framework seems reasonable to compare robustness.



**Weaknesses:**

While I like the general idea of this paper, the main thrust of it was quite difficult to follow. While I am familiar with the work "Adversarial Examples Are Not Bugs, They Are Features" (or [12]), it wasn't until nearly finishing my first read-through that the submitted manuscript is heavily based on it's work. It also wasn't clear that "tasks" in this paper do not represent different datasets (with different objectives) similar to multi-task learning, but "tasks" refers to self-supervised learning. These previous points could be made much clearer. Furthermore, it appears that the paper may have been written under time constraints, as there are multiple instances of typographical errors throughout the text ("datatset" and "hekp" for example). Taking the time to carefully proofread the content would greatly enhance its clarity and overall presentation (e.g., Fig 3 referenced on page 4, but shown on page 6). Lastly, many technical details are left out of the paper which would seriously help understanding: a brief overview of each of the SSL methods, how are the datasets constructed, and where linear probing is performed.

The proposed "theory" framework appears to be heavily based off [12], and it is more of a list of definitions rather than offering any theoretical insights. While these definitions are useful, theoretical insight could be given by analyzing how different (SSL / non SSL) tasks could effect the robustness more comprehensively.

Some of the large claims in this paper, that the robust datasets are in fact not robust, are not quite backed up by the experiments of the paper. The robust dataset was generated by using a classifier, to then train a classifier. I'm not sure why it's surprising that using that robust classifier specific dataset doesn't give robustness to SSL trained models. Here's a natural experiment:
1. Train an SSL model on the clean dataset
2. Fit a linear probe for classification
3. Use that linear probe to generate the robust dataset following [12]
4. Train a new SSL model on that new robust dataset
5. Attack that model
I would be very curious to see what Table 3 would look like after following the above steps.


Finally, results on using CIFAR-10 have questionable generalization. It is a little surprising that the "Restricted ImageNet" from [12] was not used for experiments.



**Questions:**

See Weaknesses for items to address.

**Limitations:**

See above. No ethical statement, a little could be said about dataset robustness and safety.

---

> ### Author Rebuttal · Authors · 2023-08-09
>
> #### Title: Response to reviewer nwrB
>
> We apperciate your careful reading and constructive comments on the presentation details. We will revise the paper carefully and take into consideration your suggestions in the revision.
> Below, we address your concerns on the paper content.
>
> ---
>
> **Q1.** While these definitions are useful, theoretical insight could be given by analyzing how different (SSL / non SSL) tasks could effect the robustness more comprehensively.
>
> **A1.** Thanks for suggestions. We can give an intuitive theoretical example as in Ilyas et al to illustrate the differences between SSL features and SL features.
>
> Consider a simple case where $X = USV\in\mathbb{R}^{N\times M}$ is the SVD of input features of $N$ samples with $M$-dimesional features. If there is a spurious correlation between the eigenvectors with smallest singular values and the class labels, they are useful features for classification. However, these features will be discarded in PCA that only extract top eigenvectors. This shows that the features learned by an SSL task and an SL task can be quite different. And some features may only be useful for one task only.
>
> Things are much more complex on real-world data and NN-based SSL methods. As shown in our experiments, although natural features transfer well between SSL and classification, non-robust features are mostly ineffective under various SSL training, showing strong task dependence.
>
>
> ---
> **Q2.** Some of the large claims in this paper, that the robust datasets are in fact not robust, are not quite backed up by the experiments of the paper. The robust dataset was generated by using a classifier, to then train a classifier.
>
> **A2.** We highlight that we have evaluated the robust-dataset classifier on both classification (SL->SL) and SSL (SL->SSL) tasks. The former experiment follows **the same evaluation protocol as Ilyas et al [1]'s experiments to validate the robust dataset (Sec 3.1 in their paper)**, that is to perform standard training on the robust dataset and evaluate model robustness. The *only* change we did is to replace their PGD/CW attack with AutoAttack. From Table 3 quoted below, we can find the model robustness vanishes under AutoAttack. It clearly shows that robust datasets are not really robust.
>
>
> | Attack Method | PGD-500 | CW-500 | PGD-1000 | CW-1000 | AutoAttack |
> | ----   | :----: | :----: | :----: | :----: | :----: |
> | **Robust Accuracy** | 32.86% | 32.44% | 32.59% | 32.44% | **0.21%** |
>
> > Why it's surprising that using that robust classifier specific dataset doesn't give robustness to SSL trained models? Here's a natural experiment [...]
>
> This experiment is to further study the "universal robustness" (defined in Sec 3.1) of the robust dataset, i.e., whether its robustness can generalize across different tasks. We find that similarly, SSL models using robust datasets also exhibit no robustness. In all, robust datasets show no robustness on both classification and SSL tasks.
>
> We also collect new results following the experiment setup that you suggest, using robust dataset generated by SSL (SSL -> SSL). The results are also consistent with the SL->SL and SL->SSL results, and still no nontrivial robustness is observed.
>
> | Model  |  Clean Accuracy | Robust Accuracy |
> | ----   | :----: | :----: |
> | SimCLR | 60.52% | 0.02% |
> | MAE | 28.67% | 0.24% |
> | ResNet-50 | 71.04% | 0.06%  |
> | DenseNet-121 | 72.34% | 0.10% |
>
> [1] Andrew Ilyas, Shibani Santurkar, Dimitris Tsipras, Logan Engstrom, Brandon Tran, Aleksander Madry; Adversarial Examples Are Not Bugs, They Are Features; Part of Advances in Neural Information Processing Systems 32 (NeurIPS 2019)
>
> ---
> **Q3.** Finally, results on using CIFAR-10 have questionable generalization. It is a little surprising that the "Restricted ImageNet" from [12] was not used for experiments.
>
> **A3.** Restricted ImageNet is about 1/10 size of ImageNet and extracting robust/non-robust features requires massive computation. Due to limit time of rebuttal, we instead add experiment results on TinyImageNet-200, which contains 100,000 64x64 images diveded into 200 classes.
>
> As shown in Table A in our rebuttal PDF, non-robust features still obtains much lower accuracy than natural and robust features, which aligns well with our experiments on CIFAR-10.
>
> We also evaluate the robustness of ResNet-50, DenseNet-121 with standard training on the constructed robust dataset on TinyImageNet. Simialr to CIFAR-10 results, in Table B (Rebuttal PDF), the trained models are also non-robust under AutoAttack.
>
> [1] Ya Le and Xuan S. Yang. Tiny imagenet visual recognition challenge. Stanford University. 2015
>
> [2] Andrew Ilyas, Shibani Santurkar, Dimitris Tsipras, Logan Engstrom, Brandon Tran, Aleksander Madry; Adversarial Examples Are Not Bugs, They Are Features; Part of Advances in Neural Information Processing Systems 32 (NeurIPS 2019)
>
> ---
>
> Please let us know if you have further concerns/questions.

---

> > ### Comment · Reviewer_nwrB · 2023-08-18
> > **Thanks authors**
> >
> > I'm mostly convinced by your rebuttal. I am still not a fan of the word "task" to describe different SSL losses, and I do worry about the general readability of the paper. But those are not content issues. I'll raise my score.

---

> > > ### Author Response · Authors · 2023-08-21
> > > **Thanks**
> > >
> > > Thanks for appreciating our response! We understand that the word "task" can be a bit stretched for describing different SSL objectives. We are considering changing it to be more specific, e.g., "different paradigms for representation learning". Please let us know if you have better options. Also, we will certainly carefully revise the writing to be more clear and readable, and take your valuable suggestions into consideration.

---

> ### Author Response · Authors · 2023-08-17
> **Your invaluable input is needed**
>
> Dear Reviewer nwrB, thanks for your time reviewing our paper. We have meticulously prepared a detailed response addressing the concerns you raised. Could you please have a look to see if there are further questions? Your invaluable input is greatly appreciated. Thank you once again, and we hope you have a wonderful day!

---

### Official Review · Reviewer_GnEu · 2023-07-06

**Soundness:** 3 good
**Presentation:** 4 excellent
**Contribution:** 2 fair
**Rating:** 8
**Confidence:** 4

**Summary:**

This work challenges the theory on "robust" and "non-robust" features from Ilyas et al. With an extended and more generalized formulation, the authors show that "non-robust" features are indeed very task-specific, and that even supposedly "robust" features are mostly task-specific and hardly provide robustness. The authors experiment with various self-supervised models on CIFAR10 (and its robust/non-robust variants) and answer multiple interesting research questions.

**Strengths:**

- The original work from Ilyas et al. did mention that the "robust" features are task specific, but this work explicitly tests and shows that "robust" features are not necessarily truly robust, and and only really "task-robust". In some sense, there are 2 levels of generalization- this work shows that robustness at the second stage (same X, Y distribution) does not imply robustness in the first stage (same X, different Y distribution).

- I like Eq (9) - providing a formulation that is relative to the choice of $g$.

- The conclusion in L199-200 is fitting, and ideally should have been a stress point in the original paper on robust features. The authors here have done a good job of making this point explicit to clear up the misconception around "robust" features. The fact that the original paper did claim robust features to be a property of the dataset is in direct contradiction to results from this work.

**Weaknesses:**

- My biggest concern is the interpretation of "robust" and "non-robust" features. The authors challenge the claim made by Ilyas et al. about the actual robustness of features. However, the cited paper does not claim that the "robust" features they identify are universally robust, only that they are robust for the classification task at hand. This is not surprising- 'eye color' would be a robust feature for person identification, but not for smile detection. These features are statistical patterns that ere useful for the **given** task but not of any use otherwise, so not surprising that they do not generalize to multiple tasks. Even on L112, the authors claim "we believe that the existence of non-robust features is task-reliant", which is what Ilyas et al. also say.

- L144: "...meaning that the adversarial perturbations are almost meaningless to DDPM". This can also mean that the attack is not potent enough. Also, there is no reason to believe that adversarial perturbations should transfer across tasks. The objective when adding perturbations for one task is unrelated to the wanted objective for another task- perturbations meant to fool smiling/non-smiling have no reason to influence classification scores for straight/curly hair, for instance. The former would most likely perturb areas around the face, while the latter would likely look at features close to the head.

- All experiments are focused on CIFAR-10 and its robust/non-robust variants. I would like to see at least one more dataset to be more confident in the generalization of claims made in the paper.

## Minor comments

- This paper assumes that the readers are familiar with the work on "robust features" referenced throughout the paper. Please give a brief summary of the referenced paper (from MadryLab) - their crux and notion of "robust"/"non-robust" features in the Introduction itself.

- L19: "...and it becomes natural for papers to use terms like..." - please give some examples of papers that indeed do this.

- L35: "experimente" -> "experiment"

- L41: Please provide a list of contributions. Although posing these research questions is a good way to pique interest, they should not not be left as unanswered questions until the end of the paper.

- Eq (12) seems to be missing the term $T$

- Figure 2 missing axis labels

- L164: "...exceeds 80%" - what is the ASR of a robust model for the same attack?

- L169-172: "First, the attack method.......from gradient obfuscation" - this might mean that the "robust" features are not entirely robust and have some noisy non-robust features in them too.

- Figure 4: please provide a heatmap legend. This is an interesting figure- why is cosine similarity of perturbations a good metric? Why not look at transferability rates instead. The latter would be a better and more direct indicator.

- References formatting is mixed: conference names are mixed in lower-case, capitalized, etc. For instance, "Are adversarial examples inevitable" appears in ICLR but the arxiv version is cited. Some conference names are in full while for others abbreviations are used.

**Questions:**

In Eq(11) the same $\epsilon_0$ seems to be added to added for all tasks? This does not seem optimal at all, since different tasks may require different levels of noise. For instance, a binary classifier would require much more noise (happy/sad classifier) as opposed to a more fine-grained task (person identification).

---

> ### Author Rebuttal · Authors · 2023-08-09
>
> #### Title: Response to Reviewer GnEu
>
> Dear Reviewer GnEu,
>
> Appreciate for your careful reading and acknowledging our contributions on the definitions and evaluation of robust features. Below, we address your main concerns, especially those concerning the task-reliance of robust features.
>
> ---
> **Q1.** My biggest concern is the interpretation of "robust" and "non-robust" features [...]
>
> **A1.** Thanks for your insightful comments, and we get your point that a feature could be useful for one task while useless for the other. However, this does NOT contradict with our theory. In fact, we did not expect a feature that is generalized across every possible task (e.g., eye color and smile); instead, we define robust features **for a given set of tasks $\mathcal{T}$** (see L73). The word "universally robust" could be a little stretching, but essentially, we define it as "robust for every task in $\mathcal{T}$" (see L82).
>
> For a valid discussion, in this work we consider the case when $\mathcal{T}$ contains **a set of relevant tasks where naturally trained features transfer well across different tasks**. Specifically, we consider the transfer from SSL pretraining tasks (CL, MAE, diffusion) to supervised tasks (classification), and it is a known fact that features learned by SSL are very useful for classification tasks (with high linear probing accuracy). Due to this high relevance, one would expect that like natural features, the transferability should happen for non-robust features if they are truly useful for classification (as Ilyas et al suggested). Instead, our experiments give quite surprising results that their transferability is much worse than that of natural / robust features. This supports our claim that non-robust features may not be truly useful features but mainly task-specific spurious features.
>
> We will state this relationship and make the dependence on the task set $\mathcal{T}$ clearer to avoid potential confusions. Please let us know if there is more to clarify.
>
> ---
> **Q2.** L144: "...meaning that the adversarial perturbations are almost meaningless to DDPM".
>
> **A2.** We address your concerns on this statement point by point.
>
> > This can also mean that the attack is not potent enough.
>
> We note that these non-robust features generated by PGD-1000 are very useful in the classification task, as they can 1) mislead prediction, 2) provide a good classifier when trained on non-robust dataset (Ilyas et al). The fact that it no longer works on DDPM reveals that there exits a clear discrepancy on feature usefulness between the two tasks, and thus justfies our argument here.
>
> >  Also, there is no reason to believe that adversarial perturbations should transfer across tasks.
>
> Following the discussion in A1, diffusion models can also be seen as an SSL pretrain method, and we obtain >80% transferred linear classification accuracy with diffusion-learned features (Fig 1). In view of the good transferability of natural features, the intransferability of non-robust features between two tasks reveals that non-robust features are very different from natural features.
>
> ---
> **Q3.** All experiments are focused on CIFAR-10 and its robust/non-robust variants. I would like to see at least one more dataset to be more confident in the generalization of claims made in the paper.
>
> **A3.** Within the limit of time, we reproduce some main results on a larger dataset, TinyImageNet [1], which contains 100,000 64x64 images diveded into 200 classes.
>
> As shown in Table A in our rebuttal PDF, non-robust features still obtains much lower accuracy than natural and robust features, which aligns well with our experiments on CIFAR-10.
>
> We also evaluate the robustness of ResNet-50, DenseNet-121 with standard training on the constructed robust dataset on TinyImageNet. Simialr to CIFAR-10 results, in Table B (Rebuttal PDF), the trained models are also non-robust under AutoAttack.
>
> [1] Ya Le and Xuan S. Yang. Tiny imagenet visual recognition challenge.  Stanford University. 2015
>
> ---
>
> ***Remark.* Due to the limit of characters, we address some of your key concerns in minor points below and will fix the writing problems in revision following your suggestions.
>
> **Q4.** ASR of a robust model under AutoAttack.
>
> **A4.** According to RobustBench [1], the ASR of the SOTA robust model of CIFAR10 dataset is **29.31%**, and for a medium-level adversarially trained model ASR is about **50%**.
>
> [1] Croce et al. RobustBench: a standardized adversarial robustness benchmark. NeurIPS'21.
>
> ---
> **Q5.** Figure 4: please provide a heatmap legend. This is an interesting figure-Why is cosine similarity used rather than transferability of adversarial perturbations?
>
> **A5.** We will add the legend. Consine similarity measures the relationship between adversarial perturbations in the input space. As you suggested, transferability is also a good metric for similarity and the results can be seen in Figure A in the Rebuttal PDF, which mostly consistent with previous results: the transferability between SL models is good, but poor between SSL and SL models.
>
> ---
> **Q6.** In Eq(11) the same $\epsilon_0$ added for all tasks? This might not be optimal, since different tasks may require different levels of noise.
>
> **A6.** No, $\varepsilon_0$ here denotes a sample-wise and task-wise perturabation independently generated for each sample pair $(x,y)$ at each task $T$ *per se*. Rigorously, it should be $\varepsilon_{x,\mathcal{T}}$. We will clarify the notation in revision.
>
> ---
>
> Please let us know if you have further concerns/questions.

---

> > ### Comment · Reviewer_GnEu · 2023-08-10
> > **Concerns addressed**
> >
> > The authors have addressed my concerns quite well- I will update my rating to reflect this as soon as the option becomes available. I have no further questions for the authors :) Good luck!

---

### Official Review · Reviewer_WcXm · 2023-07-08

**Soundness:** 3 good
**Presentation:** 3 good
**Contribution:** 3 good
**Rating:** 7
**Confidence:** 3

**Summary:**

In this study, the authors dispute the argument from "Adversarial examples are not bugs, they are features," where it was suggested that adversarial examples exist due to non-robust yet useful image features. They used self-supervised learning algorithms on both robust and non-robust datasets to test this theory. The findings contradicted the hypothesis, revealing that the self-supervised models trained on non-robust datasets didn't generalize well, indicating that non-robust features aren't universally useful for different training setups.

Additionally, it was demonstrated that non-robust features, while beneficial for classification, aren't helpful for reconstruction, highlighting their task-specific utility.

Moreover, models trained solely on robust features lacked robustness, showing high vulnerability to AutoAttack. This contradicts the original study's findings, as these models failed to exhibit robustness even when trained exclusively on robust data.

Lastly, an examination of cosine similarity between the adversarial attack directions of various self-supervised models showed significant differences, suggesting adversarial attacks aren't easily transferable between different training setups. This indicates that non-robust features are generally not very useful and are model-specific rather than dataset-specific.

**Strengths:**

- This paper principally examined the arguments made by [1], by training models in a self supervised manner on non-robust and robust datasets in [1].
- I think the finding the models trained on robust features alone are not robust is already a very important finding. As [1] is a paper that the community has been very interested in, it is important that other papers try to reproduce and validate the results.


[1] Ilyas, Andrew, et al. "Adversarial examples are not bugs, they are features." Advances in neural information processing systems 32 (2019).


**Weaknesses:**

- I think the authors should really expand on 6.2 and table 3. The results is truly surprising and strongly contradict the original finding in [1]. For example, the authors suggest that the use of AutoAttack is the main cause for the discrepancies with the finding of the original paper. The author should verify whether the current model is vulnerable under PGD and CW attack. If the current model is also vulnerable under PGD and CW attack with more steps, which I think the current model is going to be. Then what are the conditions for reproducing the original finding, and how does the author's setting differ from it?



**Questions:**

Could the author shows whether the current model in Table 3 is vulnerable to both PGD and CW attack? Also, could the authors ablate the experimental differences between the original paper and the current on?

**Limitations:**

The authors have addressed the limitations adequately

---

> ### Author Rebuttal · Authors · 2023-08-03
>
> #### Title: Response to Reviewer WcXm
>
> Dear Reviewer WcXm,
>
> We appreciate for your careful reading of our work as well as recognizing our finding that standard training on robust dataset does not yield true robustness. Below, we address your main concerns.
>
> ---
> **Q1.** The author should verify whether the current model is vulnerable under PGD and CW attacks. If the current model is also vulnerable under PGD and CW attack with more steps, which I think the current model is going to be. Then what are the conditions for reproducing the original finding, and how does the author's setting differ from it?
>
> **A1.** In this experiment, we follow the original setting of [1], and the only modification is to change the attacker from PGD/CW to AutoAttack. As it's shown in the table below, under the same setting, we can reproduce [1]'s results and find that the model is indeed **robust under PGD/CW attack**, even after 1000 iterations. However, under AutoAttack, that model has only 0.21% robust accuracy. Therefore, [1]'s evaluation actually gives a false sense of robustness, and their arguments on robust datasets are questionable since the resulting model is essentially non-robust.
>
> We will add this comparison to Sec 6.2 and Table 3 in revision.
>
> *Robustness of the PreActResNet-18 model obtained from standard training on robust dataset [1] under different attacks on CIFAR-10.*
> | Attack Method | PGD-500 | CW-500 | PGD-1000 | CW-1000 | AutoAttack |
> | ----   | :----: | :----: | :----: | :----: | :----: |
> | **Robust Accuracy** | 32.86% | 32.44% | 32.59% | 32.44% | **0.21%** |
>
> References:
>
> [1] Adversarial Examples Are Not Bugs, They Are Features; Andrew Ilyas, Shibani Santurkar, Dimitris Tsipras, Logan Engstrom, Brandon Tran, Aleksander Madry;  Part of Advances in Neural Information Processing Systems 32 (NeurIPS 2019)
>
> ---
>
> Please let us know if there is more to clarify. We are happy to take your further question in the discussion stage.

---

> ### Author Response · Authors · 2023-08-17
> **Your invaluable input is needed**
>
> Dear Reviewer WcXm, thanks for your time reviewing our paper. We have meticulously prepared a detailed response addressing the concerns you raised. Could you please have a look to see if there are further questions? Your invaluable input is greatly appreciated. Thank you once again, and we hope you have a wonderful day!

---

> > ### Comment · Reviewer_WcXm · 2023-08-17
> >
> > I am satisfied with the updated response, and as a result I have increased my score

---

### Author Rebuttal · Authors · 2023-08-09

The supplementary material for rebuttal.

---

### Decision · Program_Chairs · 2023-09-21

**Decision:**

Accept (poster)

**Comment:**

This paper studies the question of whether or not adversarial examples really are features or bugs.  The reviewers universally liked the paper and after an engaging discussion with the authors believed it was worth accepting. I would encourage the authors to address the potential improvements raised during the discussion period to further improve the paper.